# Exocytosis in Astrocytes

**DOI:** 10.3390/biom11091367

**Published:** 2021-09-16

**Authors:** Aleksandra Mielnicka, Piotr Michaluk

**Affiliations:** BRAINCITY, Laboratory of Neurobiology, The Nencki Institute of Experimental Biology, PAS, 02-093 Warsaw, Poland; a.mielnicka@nencki.edu.pl

**Keywords:** SNARE, secretion, vesicles, transmitter, lysosome, gliotransmission

## Abstract

Until recently, astrocytes were thought to be a part of a simple “brain glue” providing only a supporting role for neurons. However, the discoveries of the last two decades have proven astrocytes to be dynamic partners participating in brain metabolism and actively influencing communication between neurons. The means of astrocyte-neuron communication are diverse, although regulated exocytosis has received the most attention but also caused the most debate. Similar to most of eukaryotic cells, astrocytes have a complex range of vesicular organelles which can undergo exocytosis as well as intricate molecular mechanisms that regulate this process. In this review, we focus on the components needed for regulated exocytosis to occur and summarise the knowledge about experimental evidence showing its presence in astrocytes.

## 1. Introduction

Astrocytes, together with oligodendrocytes and microglia, form three main categories of glial cells in the central nervous system. They are found throughout the brain and occupy around half of the brain’s volume. Astrocytes have a bushy morphology with few main processes, starting from a relatively small cell body and extending many fine processes that invade extracellular space. It has been estimated that an astrocyte can contact over 100,000 synapses in rats and almost two million in the human cortex [1]. Even though astrocytes tend to occupy distinct and non-overlapping domains, their fine processes are in contact with one another via gap junctions. They create, therefore, a large network of interconnected cells which can easily exchange ions and small molecules of up to 1–1.2 kDa [2].

For a long time, neuroscientists considered glia as merely supportive cells in the brain that control ion and water homeostasis, produce and remove neurotransmitters, induce synaptogenesis, provide trophic factors for neurons and maintain the blood–brain barrier. Their view as “passive” cells was reinforced by observations that astrocytes expressed potassium and sodium channels and could exhibit evoked inward currents, but they did not “fire” or propagate action potentials [3]. Therefore, it can be safely said that the entire field was revolutionised by the discovery of glutamate-induced Ca^2+^ waves in cultured astrocytes. This raised the possibility that astrocytes may contribute to modulating neuronal activity [4]. This observation was later confirmed many times in different in vitro and in vivo models [5,6,7,8,9]. Further demonstrations that intracellular Ca^2+^ levels in astrocytes can cause a release of glutamate and a subsequent Ca^2+^ increase in neurons [10,11] led to the concept of gliotransmission i.e., the release of transmitters from astrocytes and other glial cells to neurons. This, in turn, helped to coin the term “tripartite synapse”, where astrocyte processes form an integral part of a functional brain connection in addition to pre- and post-synaptic compartments [12]. The classical understanding of this model assumes that through spillover neurotransmitters and other factors released by neurons, bind to high-affinity astrocytic G protein-coupled receptors (GPCRs), triggering inositol-1,4,5-trisphosphate (IP_3_) production and Ca^2+^ release from the endoplasmic reticulum (ER) [13]. Increases in astrocytic Ca^2+^ levels lead to a release of gliotransmitters, including glutamate, ATP, d-serine and γ-aminobutyric acid (GABA) [14]. The release of gliotransmitters has a wide range of effects on neurons, such as stimulating N-methyl-d-aspartate receptor (NMDAR), synchronising neuronal spiking [15], as well as regulating synaptic vesicle release probability, synaptic plasticity and even behaviour [7,14,16,17,18]. Despite this description of different modes of transmitter release from astrocytes, regulated exocytosis became the most studied but also criticised (for a review, see [7,19,20]).

Exocytosis is a process in which the cargo of a secretory vesicle is released across the cell membrane. This universal process, which is common to all eukaryotic cells, can be generally divided into two types: unregulated and regulated exocytosis. Unregulated (constitutive) exocytosis typically involves the formation of membranous secretory vesicles within the cell, in which the cargo is packaged and then continually released through the cell membrane. In regulated exocytosis, the secretory vesicle with its cargo is stored until a signal triggers the process of secretion [21,22,23]. The secretory vesicle has to go through several stages before secretion actually occurs. The first stage, docking, is the tethering and linking of the vesicle to the plasma membrane. Docking is followed by priming, during which the attached vesicle’s membrane is brought in proximity to the release site at the plasma membrane. Finally, a trigger (most commonly an influx of Ca^2+^) leads to fusion, during which the vesicle and the plasma membrane combine with each other. After fusion, the secretory vesicle can either be recycled by the closing of the fusion pore, a model termed “kiss-and-run”, or it can collapse and fully integrate with the plasma membrane [21,24]. Regardless of the mechanism of exocytosis, all cells require vesicular organelles with a cargo and the molecular machinery to conduct exocytosis.

Similar to all eukaryotic cells, astrocytes contain different types of vesicular organelle which cargo can be released to the environment. Generally, secretory vesicles form a complex network originating from the ER or the Golgi apparatus, as well as from endosomes. Astrocytes are not different, and they contain several secretory organelles, including synaptic-like microvesicles (SLMV) [25,26,27,28], dense-core vesicles (DCV) [29,30,31,32], secretory lysosomes (SL) [33,34,35] and extracellular vesicles, which can be further divided into exosomes and ectosomes [36,37]. Out of these, the SLMVs, DCVs and SLs have been well described in the literature as undergoing exocytosis in astrocytes (see Table 1). In this review, we focus on these three types of vesicular organelle, and the molecular machinery regulating exocytosis and providing its spatial organisation, as well as on the Ca^2+^ sensors which can trigger membrane fusion.

### 1.1. Synaptic-like Microvesicles (SLMVs)

Astroglial SLMVs are small (30–100 nm in dimeter) electron-lucent vesicles, similar to neuronal synaptic vesicles. They were first identified in the brain tissue in the molecular layer of the dentate gyrus using electron microscopy [25]. They are much less numerous than synaptic vesicles in neurons; however, they form small groups of 2–15 in the astrocytic cytoplasm and were shown to be present in the vicinity of NMDARs at asymmetric synapses [25,38,39]. Further studies have also found astrocytic SLMVs close to extrasynaptic NMDARs containing the GluN2B subunit [26]. Despite being grouped, SLMVs in astrocytes are not concentrated by structurally organised active zones, as it is the case in neurons [26]; however, the endoplasmic reticulum (ER) seems to be in proximity to these clusters [25,38]. This localisation is in line with the canonical tripartite synapse model, where local increases in Ca^2+^ concentration released from the ER can trigger a release of gliotransmitters. It should be noted, however, that ER is usually not present in perisynaptic processes [40]. Additionally, it was shown that SLMVs containing vesicle-associated membrane protein 2 (VAMP2) are highly mobile in astrocytes, and that an increase in cytosolic Ca^2+^ causes vesicle docking [29]. Similarly, Ca^2+^-regulated mobility of vesicular glutamate transporter 1 (VGluT1)-positive vesicles have been found also in other studies [41].

SLMVs contain mainly small signalling molecules: glutamate and D-serine [37]. Filling vesicles with a transmitter requires active transport fuelled by an electrochemical gradient generated by the vesicular ATP-dependent H^+^ pump (V-ATPase), a large complex composed of two domains, V_O_ (transmembrane, responsible for H+ translocation) and V_1_ (responsible for ATP hydrolysis) [23]. Expression of V-ATPase has been shown in cultured astrocytes by fractionation and immunoblotting [27,42], immunofluorescence [43] and functionally, by usage of its blocker—bafilomycin [44,45,46]. The transport of a neurotransmitter is mediated by vesicular transporters. There are three vesicular glutamate transporters (VGluT1-3) and they all have been identified in cultured astrocytes [25,27,29,33,34,35,45,47,48,49]. Moreover, expression of VGluT1 and -3 has been demonstrated in acute and fixed tissue by a number of techniques, including confocal and electron microscopy, and RT-PCR [25,39,46,50,51]. Electron microscopy studies have shown that VGluT1-3 colocalises with SLMVs in the hippocampus [25,39,51] and, importantly, in the case of VGluT1 and -3, immunogold labelling was not present in knockout animals of those proteins [39,51]. Additionally, Bezzi and co-workers [25], by means of immunogold staining, have shown coloclization of VGluT1 and -2 with VAMP3. Notably, other studies have not confirmed the expression of VGluTs in astrocytes by means of RNA-seq [52,53], gene chip microarrays [54], or by confocal microscopy [55]. It has to be noted, however, that high-throughput methods such as RNA sequencing or gene chip microarrays might not be sensitive enough to detect low protein expression, and that because reported exocytosis in astrocytes is slow and glutamate is also a metabolite for astrocytes, they might not need high expression of VGluTs [56,57].

So far, no d-serine vesicular transporter has been identified in astrocytes in situ; however, it was shown to be present in immunopurified astrocytic vesicles in one study [27]. The identified transporter was proposed to be the d-serine/chloride co-transporter, which uses the H^+^ gradient created by V-ATPase concentrated d-serine in the vesicles [27]. It is still not clear whether d-serine is loaded together with glutamate to the same SLMVs. This process of vesicular synergy is present in neurons where glutamate is co-released with dopamine, serotonin or acetylcholine [58]. VGluTs and d-serine are located in astrocytic VAMP2- and VAMP3-positive vesicles, which raises the possibility that these may be the same vesicles [25,45,49,59,60]. Additionally, in immunopurified SLMVs, both transmitters can be present within the same vesicle, and d-serine application modulates the uptake of glutamate to immunopurified SLMVs and vice versa [27]. This suggests that these two amino acids may be released from the same vesicles. Notably, d-serine had no effect on glutamate uptake into immunopurified synaptic vesicles [27]. Contrary to this evidence, in fixed tissue, it was shown that d-serine and glutamate are stored in distinct vesicles within the same astrocyte [38]. Notably, it has also been recently shown that the source of d-serine is mainly neuronal, as serine racemase (SR)—the enzyme which converts l-serine into d-serine—is expressed almost entirely in the neurons [61,62]. This, however, does not exclude the possibility of d-serine transport from neurons to astrocytes and secondary astroglial release of d-serine [63]. Additionally, traumatic brain injury causes a population of reactive astrocytes to express SR, so astroglial d-serine may be important in pathological conditions [64].

### 1.2. Dense-Core Vesicles (DCV)

Dense-core vesicles (DCVs) have been well studied in a variety of tissues, where they are responsible for the storage and secretion of biogenic amines, peptides and neurotrophins, including catecholamines released from adrenal chromaffin cells or insulin released from DCVs in pancreatic β cells [21]. In astrocytes, DCVs are larger than SLMVs, being 100–600 nm [30,31,32] in diameter; however, it has been reported that atrial natriuretic peptide (ANP)-containing vesicles can be 50 nm in diameter [65]. As their name suggests, DCVs have an electron-dense core, although it is not as dense as that in neuroendocrine cells [36,65]. DCVs are not very abundant in astrocytes, accounting for roughly 2% of the vesicles [29]. Despite their low numbers, they have been shown in culture to contain an array of molecules such as secretogranin II [30,31,66,67] and III [68,69], chromogranins [32], ANP [66,70,71], neuropeptide Y (NPY) [31,72], brain-derived neurotrophic factor (BDNF) [73,74] and ATP [75,76]. To our knowledge, only secretogranins consisting of DCVs have been shown to be present in situ in human brain tissue [32]. Interestingly, the same study showed the existence of inositol-1,4,5-triphosphate (IP_3_) receptors (IP_3_Rs) in DCVs, suggesting that they can serve as IP_3_-sensitive intracellular Ca^2+^ reservoirs. As they are relatively small, they could be directed to many distinct processes of the astrocyte, including perisynaptic processes [32], and provide an additional source of Ca^2+^ for regulated exocytosis. Similar to SMLVs, DCVs have also been found to be highly mobile, and an increase in cytosolic Ca^2+^ levels seems to decrease this mobility [77,78].

Of all molecules present in DCVs, ATP has attracted the most attention, as it is a potent transmitter influencing glial and neuronal signalling, as well as behaviour [79,80]. Its localisation in DCVs in cultured astrocytes was demonstrated on a few occasions [74,75,76] but, to our knowledge, not in the tissue. ATP is transported into vesicles by the vesicular nucleotide transporter (VNuT) [81], which is also present in microglia [82]. VNuT is present in cultured astrocytes [47,83,84,85] as well as in freshly isolated astrocytes [86]. Even though there seems to be a lack of evidence showing VNuT in the astrocytes in tissue, an astrocyte-selective VNuT knockout was shown to be important for fluoxetine-induced antidepressive behaviour [87]. This provides proof, albeit indirect, that ATP is transported into the vesicles in astrocytes in vivo.

### 1.3. Secretory Lysosomes (SL)

Secretory lysosomes have diameters ranging from 300 to 500 nm [34,88] and have been identified multiple times in cultured astrocytes or in freshly isolated astrocytes, where they store and release ATP in a Ca^2+^-dependent manner [34,35,89]. Moreover, lysosomes can coexist with SLMVs in the same astrocyte, and they fuse with the plasma membrane in a Ca^2+^-regulated manner, although small vesicles are exocytosed more efficiently than lysosomes [33]. Secretory lysosomes contain ATP, cathepsin B and D and other proteolytic enzymes [34,37,90,91]. Secretory lysosomes in cultured astrocytes can be labelled by dextrans [92,93], various FM dyes and MANT-ATP—a fluorescent analogue used in studies of ATP stores [34,94]—as well as quinacrine, which emits a green fluorescent signal in the presence of intracellular ATP [91]. One study has found, however, that FM dyes do not enter astrocytes on the endocytic pathway but rather via the store-operated calcium channel and do not stain lysosomes [95]. In this study, the authors did not use any lysosomal/endosome markers except for fluorescently labelled dextran. Astroglial secretory lysosomes lack VGluTs, VAMP2 and VAMP3 [33,34]; however, they express lysosomal-specific markers including cathepsin D, lysosomal-associated membrane protein 1 (LAMP1) [34,59], sialin, CD63/LAMP3 [35], monomeric ras-related protein Rab7 [96] and VAMP7 [35]. Interestingly, VAMP7 is insensitive to cleavage by the tetanus neurotoxin (TeNT) and contributes to TeNT-independent exocytotic release of ATP, hence its alternative name (TI-VAMP) [33,91]. Downregulation of VAMP7 expression inhibits the fusion of ATP-storing vesicles and decreases ATP-mediated calcium wave propagation [91], which is an important form of long-range communication in the astrocytic network. Similar to DCV, secretory lysosomes in the astrocytes express VNuT [83], which transports ATP from the cytoplasm into the vesicles.

## 2. The Exocytosis Machinery in Astrocytes

Secretory vesicle docking, priming and fusion with a plasma membrane are generally mediated by SNARE proteins (soluble N-ethylmaleimide-sensitive fusion protein attachment protein receptors) and SM proteins (Sec1/Munc18-like proteins) that undergo a cycle of association and dissociation during the fusion reaction. The mechanism of regulated SNARE complex assembly is conserved in many different cell types, including neuronal, exocrine, haematopoetic and endocrine cells (Figure 1; for reviews, see [21,23,97,98,99]). The SNARE complex consists of two target membrane proteins (t-SNARE), namely syntaxin and SNAP-25 (or SNAP-23), and one vesicle-associated (v-SNARE) protein, VAMP2. Syntaxins are ~35 kDa proteins containing a carboxy-terminal transmembrane domain and an amino terminus oriented toward the cytoplasm. The other t-SNARE, SNAP-25 or -23 (25 and 23 kDa respectively), is attached to the plasma membrane via four palmitoylated cysteine residues. The v-SNARE VAMP2 is a 18 kDa protein with a vesicle lumen-oriented carboxyl terminus and an amino terminus oriented toward the cytoplasm. During the vesicle docking process, the SM protein Munc18-1 initially binds to the closed conformation of syntaxin1. When the closed conformation of syntaxin1 ‘‘opens’’ during priming and SNARE complexes starts forming, Munc18-1 remains attached to syntaxin1 in the assembling SNARE complex but switches its binding mode to an interaction with the SNARE complex. The SNARE complex is extremely stable, where one v-SNARE binds with two t-SNARE proteins in a 1:1:1 ratio. The SNARE complex forms initially a “*trans*” configuration (SNARE proteins are on opposite membranes) with amino- to carboxy-terminal zippering of proteins, which brings the vesicle closer to the plasma membrane. Assembly of the full *trans*-SNARE complex, together with the action of the SM protein, opens the fusion pore. After fusion pore opening, the membrane of the vesicle and the plasma membrane completely merge, and *trans*-SNARE complexes are converted into *cis*-SNARE complexes, where the SNARE complexes are on a single membrane. Next, the *cis*-SNARE complex proteins are bound by N-ethylmaleimide sensitive factor (NSF) and soluble NSF-attachment proteins (SNAPs, no relation to SNAP-25 and its homologs) to catalyse SNARE complex dissociation into monomers. This allows for endocytosis of the v-SNARE and recycling of the individual t-SNAREs back to their respective plasma membrane compartments.

For over two decades, a large number of studies identified components of the SNARE complex and SM proteins in the astrocytes; however, the issue of regulated exocytosis still remains a matter of debate. Out of the v-SNAREs, the expression of VAMP2 and VAMP3 has been shown multiple times in vitro [29,45,100,101,102] and in vivo [43,46,52,53,54,103].

The functionality of VAMP2/VAMP3 has been confirmed by multiple studies, especially by the use of tetanus neurotoxin (TeNT) and botulinum neurotoxin (BoNT), which cleave SNARE proteins. In particular, application of TeNT to cultured astrocytes attenuated the exocytotic release of glutamate [25,45,104,105,106]. Moreover, the enzymatically active light chain of TeNT was applied through a patch pipette to astrocytes in acute slices, where it abolished the release of d-serine [16] or glutamate [26,107], which affected neighbouring neurons. Importantly, Haydon’s laboratory has developed a transgenic mouse which overexpress a dominant negative SNARE (cytosolic tail of VAMP2, dnSNARE) in astrocytes [17]. This method had been previously shown to decrease astrocytic gliotransmission in vitro [46] and was later explained in detail as stabilizing the fusion pore in a narrow, release-unproductive state, thus effectively decreasing exocytosis [73]. Experiments with dnSNARE transgenic mice performed not only by Haydon’s group showed reduced SNARE-dependent gliotransmission (mostly of adenosine and ATP), which influences the behaviour, synaptic transmission and maturation of neurons [17,86,108,109,110,111,112,113]. Although studies using dnSNARE mice were convincing and showed the transgenic expression specifically in astrocytes, a study by Fujita and co-workers [114] found that dnSNARE can be also expressed in the neurons of transgenic mice, thus questioning the previous findings. This has led to a heated debate (see [115,116]). The discussion has pointed to technical matters in studying gliotransmission, particularly the use of the GFAP promoter as a glial-specific promoter. Importantly, another study using astrocyte-specific expression of the BoNT serotype B light chain (BoNT/B) in a Cre/Lox system (named “iBot mice”) showed decreased VAMP-dependent glutamate release from the astrocytes and impaired glial volume regulation [117]. Later, the same model (iBot mice) was used in parallel with dnSNARE mice to show that blockade of gliotransmitter release from astrocytes influences adult-born neurons, reduces their glutamatergic synaptic input as well as dendritic spine density, and leads to their lower functional integration and hence survival [110]. This result validated the relevance of the dnSNARE mouse model, as well as the importance of astrocytic exocytosis.

Other v-SNARE proteins have also been identified in astrocytes. In particular, VAMP7 was shown in astrocytes in vitro as a v-SNARE associated with lysosomes [35,59,91], as was VAMP8 [35,59]. Additionally, VAMP4 and VAMP8 can be found to be actively expressed in vivo in the cortex, hippocampus and striatal astrocytes, based on RNA sequencing [53].

For t-SNARE proteins, astrocytes were shown to express syntaxin1; however, the results of various studies have been mixed. Syntaxin1 has been demonstrated in cultured astrocytes [45,66], albeit at a lower level than in isolated synaptosomal membranes [118]. Additionally, one group failed to find syntaxin1 in cultured astrocytes [119], while another was able to identify it in astrocytes in situ [120]. Syntaxin4, however, seems to have a much higher expression level in astrocytes than syntaxin1, and it was shown to be expressed in culture [66], in situ [121] and ex vivo from freshly isolated astrocytes, based on RNA sequencing [53]. With regard to the second t-SNARE, it seems that astrocytes express SNAP-23 rather than SNAP-25. The latter failed to be found even in cultured astrocytes [45,66,118,119,122,123]. SNAP-23 was, however, identified in astrocytes in vitro [27,29,43,45,46,48,66,122,123] and in fixed tissue in vivo [43,120], although one recent study has found its expression to be insignificant, based on RNA sequencing, while showing SNAP-25 expression [53]. Therefore, the expression profile of SNARE proteins in astrocytes seems to follow other non-neuronal cells, where the SNARE complex consists mostly of SNAP-23, syntaxin 4 and VAMP2 (synaptobrevin 2) or VAMP3 (cellubrevin) [124,125,126].

As mentioned above, the SNARE complex requires SM proteins for its formation, in particular Munc-18 is a prominent and well-studied example. It was shown to be significant for vesicle docking in chromaffin cells where Munc18-1 knockout reduces fraction of docked vesicles from 30% to 5% [21,127]. Moreover, Munc18-1 binding to assembling SNARE complexes is essential for synaptic vesicle fusion, and in every SNARE-dependent fusion reaction studied, an SM protein participated and was essential for that fusion reaction [98]. Munc18-1 and Munc18-3 have been found in cultured and freshly isolated astrocytes [46,66,128], while RNA sequencing from freshly isolated astrocytes has shown that they express Munc18-3 rather than Munc18-1, which is expressed in the neurons [52]. A similar result has recently been found in another study using RNA sequencing, where Munc18-1 expression was found in astrocytes at a low level; however, Munc18-3 and Munc18-4 were enriched in the astrocytes [53].

Other components of the exocytosis machinery in astrocytes have not been intensively studied. Notable exceptions are the Rab proteins—small GTPases that are localised on synaptic vesicles and are important for their docking. Cultured astrocytes were shown to express Rab3 [100,101,129,130], which is usually associated with SMLVs or DCVs. A recent study in cultured astrocytes has shown that disruption of Rab3 function by mutated Huntingtin blocks the docking of DCVs and ultimately decreases BDNF and ATP release [74]. Expression of Rab7, which is associated with late endosomes/lysosomes, was also reported in astrocytes in vitro [29,35,91,131]. Additionally, Rab10 and Rab35 have also been shown to be associated with lysosomes in cultured astrocytes [131]. Thanks to access to a database (http://astrocyternaseq.org/ accessed on 1 January 2021) of sequenced astrocytic RNAs from different brain structures [53], we were able to confirm the expression of Rab3a, Rab7 and Rab35 in freshly isolated astrocytes. All these proteins are expressed at a significantly different level from the expression threshold set as fragments per kilobase per million (FPKM) >10 (see [53]).

## 3. Spatial Organisation of Exocytosis

Ultrafast neurotransmitter release, which occurs in neurons in response to an action potential, can only be achieved by bringing Ca^2+^ channels to docked and primed synaptic vesicles at the active zone. Most of the active zones in vertebrates are specialised disc-like structures at the plasma membrane with a 0.2–0.5 μm diameter. Active zones are surrounded by a perisynaptic zone containing transsynaptic cell adhesion molecules and receptors regulating neurotransmitter release, which is the site of synaptic vesicle endocytosis [132]. The core of the active zone is formed mainly by the members of six evolutionarily conserved proteins: Rab3-interacting molecules (RIM), RIM-binding proteins (RIM-BP), Munc13 (no relation to Munc18), α-liprin and ELKS protein, as well as piccolo/bassoon. RIM, RIM-BP and Munc13, are multidomain proteins, whereas α-liprin and ELKS exhibit a simpler structure. Active zone proteins form a single large protein complex that docks and primes synaptic vesicles, recruits Ca^2+^ channels to the docked and primed vesicles, tethers the vesicles and Ca^2+^ channels to synaptic cell adhesion molecules, and mediates synaptic plasticity. Astrocytes do not have an active zone that could be observed under an electron microscope; however, the fact that there have been observations of small groups of SMLVs in astrocytes (see above) [25,38,39] suggest the possibility of its existence. To our best knowledge, components of the active zone have not been intensively studied, except for Munc13, which is a priming factor catalysing the conformational switch of syntaxin1 from closed to open, promoting SNARE complex assembly. Mungenast [133] found the expression of Munc13-1 in cultured astrocytes by RT-PCR, immunostaining and Western blotting, and showed that siRNA downregulation of Munc13-1 inhibits astrocytic ATP release, induced by diacylglycerol (DAG). In another study based on RNA sequencing from freshly isolated astrocytes, Munc13-2 and -3 were enriched in the astrocytes, while Munc13-1 was expressed mainly by neurons [52,134]. Similar results can be found in RNA sequencing data from freshly isolated astrocytes; however, only Munc13-3 has shown a significant level of expression in these cells in the striatum [53]. Results showing the expression and function of Munc13 in regulated exocytosis in the astrocytes are very limited, and the involvement of other active zone components in vesicular release could provide more insights into explaining the process. It can be theorised, however, that the spatial organisation of exocytosis in astrocytes might be completely different from that present at a presynaptic site in neurons. For example, a recent report by Buscemi and co-workers showed that mGlu5 interacts with Homer1b/c in astrocytes and that this interaction is crucial for the their glutamate release [135]. The interaction between Homer and mGluR5 in astrocytes has been previously suggested but not shown directly [136]. Homer1 is, however, a neuronal postsynaptic marker (for a review, see [137]) and its role in astrocytes is quite novel. Nevertheless, in line with these observations, neurons may also have “non-canonical” exocytosis machinery, which has been reported, for example, in dendrites [138]. Interestingly, the composition of the SNARE complex, which was proposed to regulate exocytosis from neuronal dendrites, is very similar to that proposed in astrocytes and consists of SNAP-23, syntaxin 4 and VAMP2 or VAMP3 [138].

## 4. The Ca^2+^ Sensor for Regulated Exocytosis in Astrocytes

Under physiological circumstances, primed vesicles are stimulated to exocytosis by Ca^2+^; therefore, to trigger the final stage of the fusion reaction, a Ca^2+^ sensor is required at the site of exocytosis. Since the discovery of synaptotagmin1 (Syt1) [139], it has been proposed to be the Ca^2+^ sensor for regulated exocytosis. Syts are evolutionarily conserved proteins containing a short amino-terminal sequence directed toward the vesicle lumen, followed by a transmembrane region, a central linker sequence of variable length, and two carboxy-terminal C2 domains that bind Ca^2+^. [98,140]. C2 domains were initially defined in protein-kinase C isozymes and were later shown to constitute autonomously folding Ca^2+^/phospholipid-binding domains. In addition, C2 domains constitute protein interaction domains and, in the case of Syt1, bind to syntaxin-1 and to SNARE complexes. There are 16 Syts expressed in the brain and eight of them bind Ca^2+^: Syt1, Syt2, Syt3, Syt5, Syt6, Syt7, Syt9 and Syt10 [140]. Out of those, Syt1, Syt2 and Syt9 are responsible for triggering fast fusion of the synaptic vesicles; however, astrocytes do not seem to express any of them [52,53,123,141,142]. Strikingly, for many other forms of regulated, Ca^2+^-triggered exocytosis, Syt1, Syt2 and Syt9 are involved in chromaffin cells, neuropeptide secretion in neurons or mast cell degranulation. However, Syt7 has also been shown to be involved in triggering exocytosis in chromaffin cells [143], pancreatic insulin- and glucagon-secreting cells [144,145], or exocytosis of lysosomes in fibroblasts [146]. In neurons, Syt7 was also shown to be important for Ca^2+^-triggered asynchronous release [147]. For astrocytes, however, results showing Syt7 expression are quite ambiguous. Studies in cultured astrocytes did not detect Syt7 [142,148] or even showed that overexpression of Syt7 inhibited lysosomal exocytosis [148]. However, Mittelsteadt and colleagues [141] showed, using single cell RT-PCR, that patch-clamped hippocampal astrocytes in the CA1 stratum radiatum express Syt7 (in seven out of nine cells). However, cultured astrocytes express Syt4 and Syt11 [29,65,123,142,148,149,150]. Syt4 expression was also found in freshly isolated astrocytes by RT-PCR [142]; similarly, Syt11 was found by RNA sequencing [52,53] and RT-PCR [141]. Zhang and co-workers also confirmed the expression of Syt4 in situ by confocal and electron microscopy. Even though Syt4 and Syt11 do not bind Ca^2+^ [151], Zhang and co-workers have shown that it regulates astrocytic Ca^2+^-dependent glutamate release in cultured astrocytes [142]. Similarly, Syt11 was shown to regulate Ca^2+^-dependent lysosome exocytosis in injured astrocytes [148]. It should be mentioned that there is a whole set of studies showing that astrocytes do not release neurotransmitters in response to Ca^2+^ elevation, arguing against the existence of Ca^2+^-dependent exocytosis in astrocytes (for in depth reviews, see [7,19,20,115,152]). Clearly, the involvement of synaptotagmins in regulated exocytosis in astrocytes needs further studies and clarification, especially in more intact preparations like brain slices or in vivo.

Synaptotagmins, however, do not act alone in triggering fusion but require complexin as a cofactor [140]. Complexin was discovered by virtue of its tight binding to SNARE complexes. It functions as a priming factor for SNARE complexes, as well as an activator of these complexes, preparing them for subsequent synaptotagmin action, and also as a clamp of spontaneous release, preventing unregulated fusion [98]. Complexin’s expression in astrocytes has been studied mainly in vitro, where cultured astrocytes have been shown to express complexin 2, while complexin 1 was expressed in the neurons [46,153,154]. Recently, complexin 2 expression was confirmed by RNA sequencing data [53], while complexin 1 was also expressed at a significant level, albeit much lower than the general tissue level, and only in cortical astrocytes.

It should also be noted that Ca^2+^ is not the only signal that can trigger vesicle secretion, as many cells show GTP-dependent DCV secretion. Non-hydrolysable GTP was shown to trigger secretion in a Ca^2+^-independent manner in chromaffin cells [155], mast cells [156] and pancreatic β cells [157]. Even though GTP-dependent exocytosis differs from Ca^2+^-dependent exocytosis in terms of the signal sensor for triggering, the final fusion steps are still dependent on the SNARE proteins [21]. The major sensors for GTP in GTP-dependent exocytosis are considered to be Ral proteins (RalA and RalB) and GTPases [158]. Ral proteins interact with the exocyst protein complex, which is an octameric protein complex containing, among other proteins, Sec5, which is bound by Ral in a GTP-dependent manner [21,159]. It is believed that the exocyst complex tethers secretory vesicles to the plasma membrane, where fusion occurs in a SNARE-dependent manner [159]. For example, live-cell imaging of Sec8 (another member of the exocyst complex) showed that it is transported to a cellular membrane where it remains for seconds until fusion occurs [160], which is in line with the slow kinetics of astrocytic release after stimulus [161]. Additionally, members of the exocyst complex can interact with Rab proteins or v-SNARE [162], directly with plasma membrane via phosphatidylinositol 4,5-bisphosphate (PI(4,5)P2) [163], and with a Sec1/Munc18 (SM) protein family member, Sec1 [164]. Interestingly, when we searched for expression of RalA, RalB and Sec5 (encoded in mice by Ral1a, Ral1b and Exoc2, respectively) in a database of sequenced astrocytic RNAs (http://astrocyternaseq.org/ accessed on 1 January 2021), we found that all these proteins are expressed at a significantly different level from the expression threshold set as fragments per kilobase per million (FPKM) >10 [53]. However, to our knowledge, so far, there are no experimental data exploring the exocyst complex in astrocytes, as it has mostly been studied in yeast. Given the lack of Ca^2+^-binding synaptotagmins in astrocytes and their slow exocytosis [37,161], GTP-dependent exocytosis could be an interesting alternative hypothesis.

## 5. Exocytosis in Pathological Conditions

In nearly all brain pathologies, such as traumatic brain injury, stroke, ischemia, infectious disease, neuroinflammatory and neurodegenerative disease, epilepsy, brain tumours, schizophrenia, migraine or depression, there is clearly a presence of “reactive astrogliosis” [165,166]. This term is a result of a consensus statement [167] recently proposed to describe the process in which, in response to pathology, astrocytes engage in molecularly defined programs involving changes in transcriptional regulation, biochemical, morphological, metabolic and physiological remodelling, which result in a loss of or increase in homeostatic functions and/or the gain of completely new function(s).

Accumulating evidence suggests that in pathological conditions, microglia are activated first and, through the release of ATP and inflammatory mediators (mainly interleukin-1α (IL-1α), tumour necrosis factor (TNF) and complement component 1, subcomponent q (C1q)), subsequently trigger astrocytic activation [168,169]. Activated astrocytes, in turn, increase their production and secretion of chemotactic cytokine stromal cell-derived factor-1α (SDF-1α) [170,171], the proinflammatory cytokine TNF and the inflammatory mediator prostaglandin E2 (PGE2) [105,172], IL-1α, IL-6, interferon-γ (IFN-γ) [173] and many others [174]. As mentioned above, some groups have argued that astrocytes are capable of Ca^2+^-regulated exocytosis only after activation by microglia (see, for example, [115,175]). In support of this view, a previous report by Pascual and co-workers [176] showed that activation of the microglia by lipopolysaccharide (LPS) induced a rapid (within minutes) and transient (several minutes long) increase in the frequency of excitatory synaptic currents in acute hippocampal slices. The proposed mechanism involved the activation of metabotropic P2Y1 receptors (P2Y1Rs) on astrocytes by the release of ATP from the microglia, thus triggering glutamate release from astrocytes and finally modulating synaptic mGluRs. Similarly, it was shown that astrocytes from TNF-deficient or TNF type 1 receptor (TNFR1)-deficient mice displayed altered P2Y1R-dependent Ca^2+^ signalling and decreased glutamate release [44,177]. Interestingly, a recent study [178] has shown that the presence of microglia downregulates the expression of VAMP2 in astrocytes, which slows down the vesicular release by astrocytes. This favours the longer release of transmitters in opposition to a rapid release and exhaustion of the vesicular pool in the absence of microglial factors. Additionally, the authors showed that the gliotransmission triggered by the P2Y1 agonist is impaired in slices from transgenic mice devoid of microglia [178].

SDF-1α, TNF or PGE2 by themselves are also sufficient to induce glutamate release from astrocytes, albeit still in a Ca^2+^-dependent manner. They act through activation of their respective receptors—CXCR4 and-, TNFR (both G_i/o_–associated GPCRs)—or prostaglandin E (EP) receptors (G_i_/G_s_–associated GPCRs) [104,105,175,179,180,181]. The Ca^2+^-dependence of the exocytosis of glutamate after activation of these receptors is based on an observation that the response is blocked by intracellular Ca^2+^ chelators or inhibitors of exocytosis [105].

In physiological conditions, astrocytes maintain constant levels of blood and brain *P*CO_2_/pH and they respond to physiological decreases in pH with vigorous elevations in intracellular Ca^2+^ and regulate the exocytosis of ATP [182]. However, in pathological conditions of hypoxia, they can react to a decrease in *P*O2 by Ca^2+^ signalling, accompanied by increases in mitochondrial reactive oxygen species (ROS) production and ATP exocytosis [183,184]. In the context of hypoxia, a recent study by Byts and co-workers [185] has shown that transmembrane prolyl 4-hydroxylase (P4H-TM), which is located in the ER and has a Ca^2+^-sensing EF-domain, controls ATP-induced Ca^2+^ signalling and gliotransmission. This effect is mediated by hypoxia-inducible factor 1 (HIF1), which is a heterodimeric transcription factor. In normoxia, prolyl 4-hydroxylase (P4H) hydroxylates two prolyl residues located on HIF’s α subunit. This hydroxylation leads to von Hippel–Lindau (VHL)-targeted degradation of HIFa, which suppresses the transcription of hypoxia-responsive genes. However, in hypoxia, P4H is inactive, which leads to stabilisation, accumulation and activation of HIF and induction of hypoxia-responsive genes. One study [185] showed that P4H-TM knockout animals had a changed expression of several genes related to Ca^2+^ signalling and vesicular transport/docking pathways, thus influencing receptor-operated calcium entry (ROCE) and store-operated calcium entry (SOCE), as well as calcium re-uptake by mitochondria. Moreover, in an in vitro model of cerebral ischemia, oxygen–glucose deprivation (OGD), there was a biphasic increase in Ca^2+^ signalling in astrocytes as well as neurons and a subsequent accumulation of pro-inflammatory factors, such as IL-1b and TNFα, leading to hyperexcitation of the neurons and their death after reoxygenation [186]. Interestingly, pretreatment of cell cultures with the selective α2-adrenergic receptor agonists guanfacine and UK-14,304 showed a neuroprotective effect through Ca^2+^-regulated exocytosis of ATP [186].

Reactive astrocytes have been also shown to release complement system proteins [187]. The complement system represents one of the most basic immune cascades. The complement proteins C3a, C1q and C5 are present in the brain, where they regulate neurogenesis, neuronal survival and synaptic elimination [188,189]. It has been shown that NF-κB signalling promotes the secretion of C3a. In Alzheimer’s disease, exposure to amyloid β strongly activates astroglial NF-κB, which increases the astroglial C3a release that, in turn, contributes to neurodegeneration [187].

Finally, the activation of astrocytes is often associated with their morphological and biochemical remodelling. The reactivity is manifested by an increased expression of intermediate filaments (most notably GFAP and vimentin). It has been suggested that such an upregulation of intermediate filaments allows faster and therefore more efficient delivery of major histocompatibility complex (MHC) Class II molecules to the cell surface. Exposure of astrocytes to INF-γ induced MHC Class II expression in late endosomes/lysosomes [92,96].

## 6. Conclusions

Astrocytes are receiving increasing attention as their complex role in the central nervous system becomes more prominent and accepted. In this review, we presented only a fraction of the studies which not only show that astrocytes have various types of secretory vesicles but also that they express a complex molecular machinery associated with those vesicles which is sufficient for their regulated exocytosis. For other excellent in-depth reviews, please see [36,134,161]. Even though some of the components of this release machinery, such as SNARE or synaptotagmins, have been intensively studied for over two decades, others have remained mostly undescribed. These include potential scaffolding proteins or components of the active zone, which can direct exocytosis to the most crucial places for astrocyte–neuron communication, but also alternative to Ca^2+^ mechanisms of triggering exocytosis. Moreover, much attention needs to be given to designing new experiments to bring them to the more physiology-relevant environment of neural tissue. New techniques of astrocyte culturing also give hope for creating new study models which will resemble astrocytes in the brain. Regardless of the discussion on Ca^2+^ sources for triggering exocytosis and gaps in our knowledge about the elements of the exocytotic molecular machinery, astrocytes have proven to be an important component of neuronal networks.

## Figures and Tables

**Figure 1 biomolecules-11-01367-f001:**
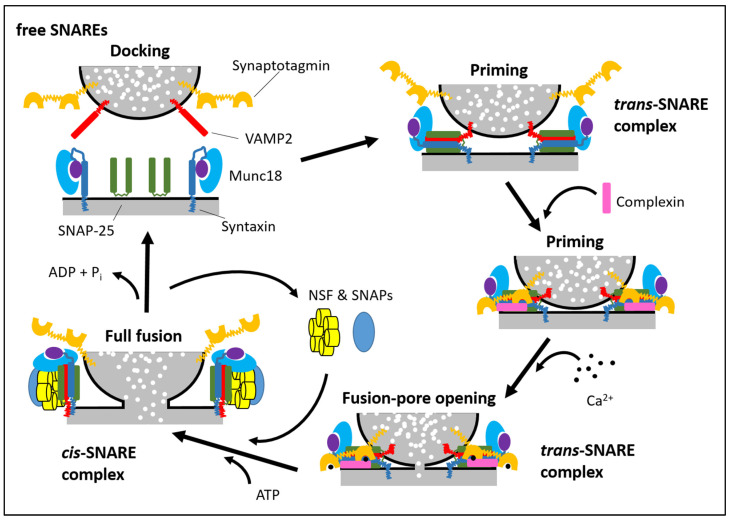
Schematic SNARE/SM cycle. SNARE and SM proteins undergo a cycle of assembly and disassembly. At the beginning of docking, syntaxin1 is present in a “closed” conformation in which its Habc domain (purple) blocks its SNARE motif (dark blue rectangle). In this position, Munc18-1 binds monomeric syntaxin1. For the SNARE complex to assemble, syntaxin1 has to ‘‘open’’. During this conformational change, the SNARE complex assembly and Munc18-1 change their binding to syntaxin1 by binding to assemble the *trans*-SNARE complexes via interacting with the syntaxin1 N-peptide. Once the SNARE complexes have partly assembled, complexin binds to further tighten secretory vesicle priming. The ‘‘superprimed’’ SNARE/SM protein complexes are then ready for the Ca^2+^-trigger. Ca^2+^ binds to synaptotagmin, which causes an interaction between synaptotagmin and SNAREs and phospholipids of the plasma membrane. After fusion pore opening, the vesicular membrane and plasma membrane merge, resulting in a change from *trans*- to *cis*-SNARE complexes. The association of NSF/SNAP ATPases disassembles SNARE complexes to free SNAREs, and the vesicle is recycled, can be refilled with neurotransmitters, and reused for another release (modified from [98]).

**Table 1 biomolecules-11-01367-t001:** Secretory vesicles undergoing exocytosis in astrocytes.

Secretory Organelle	Diameter	Cargo	Associated Proteins	
			Protein Name	Gene Name
Synaptic-Like Microvesicles (SLMVs)	30–100 nm	Glutamate d-serine	VGluT1 VGlut2 VGlut3 VAMP2 (synaptobrevin 2) VAMP3 (cellubrevin) Rab3a V-ATPase	Slc17a7 Slc17a6 Slc17a8 Vamp2 Vamp3 Rab3a Atp6v0, Atp6v1 ^1^
Dense-Core Vesicles (DCV)	100–600 nm	ANP ATP BDNF Secretogranin II Secretogranin III Chromogranin NPY	VAMP2 (synaptobrevin 2) VAMP3 (cellubrevin) VNuT	Vamp2 Vamp3 Slc17a9
Secretory Lysosome (SL)	300–500 nm	ATP Cathepsin B Cathepsin D Proteolytic enzymes	VAMP7 (TI-VAMP) Rab7 CD63 LAMP1 Sialin VNuT	Vamp7 Rab7a Cd63 Lamp1 Slc17a5 Slc17a9

^1^ V-ATPase is a large complex consisting of 13 subunits coded by many more genes; however, all the gene names start with Atp6v0 or with Atp6v1 for the V_O_ and V_1_ domains, respectively.

## Data Availability

Not applicable.

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
