# Peer review of "Exocytosis in Astrocytes"

_biomolecules, 2021, doi:10.3390/biom11091367_

Round 1
Reviewer 1 Report
Dear Authors,
The manuscript titled "Exocytosis in astrocytes" is very interisting, well written and logically constructed.
I have just few question and suggestion in order to improve your work.
Indeed, astrocytes as well as other glial cells are extremely important both during physiological and pathological condition. In particular, astrocytes play a pivotal role during inflammatory status. As recently quoted (Nosi et al., 2021: 10.3390/cells10051195) the astrocytes have different status during inflammation as well as microglia. On this regard, it should be fine and appropriate if you can include a subheading mentioning what changes occur during this status.
Author Response
Dear Reviewer,
Following your suggestion, we have added an appropriate section concerning astrocytes exocytosis in pathological conditions.
With kind regards,
Piotr Michaluk
Reviewer 2 Report
The review is written consistently and in detail, the authors clearly state the latest data on the topic of exocytosis in astrocytes. This review is of significant interest and is relevant, but the latest data on new information about this topic cannot be ignored. For example, it would be very interesting and useful to supplement the review with information about the new regulator of calcium signaling in astrocytes, Prolyl 4-Hydroxylase.
And some information about the neuroprotective mechanisms of astrocytes in OGD/reoxygenation conditions can be added
https://www.researchgate.net/publication/337839964_A_Complex_Neuroprotective_Effect_of_Alpha-2-Adrenergic_Receptor_Agonists_in_a_Model_of_Cerebral_Ischemia-Reoxygenation_In_Vitro
https://pubmed.ncbi.nlm.nih.gov/33114758/
https://doi.org/10.3390/ijms22168805
https://science.sciencemag.org/content/329/5991/571
Optional, but good experimantal and theoretical data
https://pubmed.ncbi.nlm.nih.gov/29274121/
https://pubmed.ncbi.nlm.nih.gov/26203141/
Author Response
Dear Reviewer,
1) it would be very interesting and useful to supplement the review with information about the new regulator of calcium signaling in astrocytes, Prolyl 4-Hydroxylase.
We added the suggested information to the manuscript
2) And some information about the neuroprotective mechanisms of astrocytes in OGD/reoxygenation conditions can be added
https://www.researchgate.net/publication/337839964_A_Complex_Neuroprotective_Effect_of_Alpha-2-Adrenergic_Receptor_Agonists_in_a_Model_of_Cerebral_Ischemia-Reoxygenation_In_Vitro
We have added the suggested information to the manuscript
3) https://pubmed.ncbi.nlm.nih.gov/33114758/
We can't find the relation of the above publication to the submitted manuscript concerning exocytosis in astrocytes. The publication concerns modelling of experimental data of Ca2+ imaging in response to ATP stimulation and in ischemic conditions. There are no suggestions that the observed effects can be due to exocytosis. We have already acknowledged in the manuscript exocytotic release of ATP or inflammatory molecules from astrocytes on multiple occasion in the manuscript. We have added the whole section concerning exocytosis from astrocytes in pathological conditions.
4) https://doi.org/10.3390/ijms22168805
Again, we can't find the relation of the above publication to the topic of the manuscript. The publication concerns the effects of antioxidant enzyme peroxiredoxin-6 (Prx-6) in ischaemia. Exogenous Prx-6 has a complex protective effect on hippocampal cells during ischemia with tendency towards more selective protection of astrocytes. Incubation with Prx-6 changes the basic antioxidant status of cells through changes in the expression of key genes, resulting in the suppression of ROS production by mitochondria under OGD conditions.
5) https://science.sciencemag.org/content/329/5991/571
We have added suggested information to the manuscript
6) https://pubmed.ncbi.nlm.nih.gov/29274121/
We have added suggested information to the manuscript
7) https://pubmed.ncbi.nlm.nih.gov/26203141/
We have added suggested information to the manuscript
With best regards,
Piotr Michaluk